# Pareto Adversarial Robustness: Balancing Spatial Robustness and Sensitivity-based Robustness

## Abstract

Adversarial robustness, mainly including sensitivity-based robustness and spatial robustness, plays an integral part in the robust generalization. In this paper, we endeavor to design strategies to achieve comprehensive adversarial robustness. To hit this target, firstly we investigate the less-studied spatial robustness and then integrate existing spatial robustness methods by incorporating both local and global spatial vulnerability into one spatial attack design. Based on this exploration, we further present a comprehensive relationship between natural accuracy, sensitivity-based and different spatial robustness, supported by the strong evidence from the perspective of representation. More importantly, in order to balance these mutual impact within different robustness into one unified framework, we incorporate the Pareto criterion into the adversarial robustness analysis, yielding a novel strategy towards comprehensive robustness called *Pareto Adversarial Training*. The resulting Pareto front, the set of optimal solutions, provides the set of optimal balance among natural accuracy and different adversarial robustness, shedding light on solutions towards comprehensive robustness in the future. To the best of our knowledge, we are the first to consider comprehensive robustness via the multi-objective optimization.

## 1 Introduction

Robust generalization can serve as an extension of tradition generalization, i.e., Empirical Risk Minimization in the case of i.i.d. data (Vapnik & Chervonenkis, 2015), where the test environments might differ slightly or dramatically from the training environment (Krueger et al., 2020). Improving the robustness of deep neural networks has been one of the crucial research topics, with various different threads of research, including adversarial robustness (Goodfellow et al., 2014; Szegedy et al., 2013), non-adversarial robustness (Hendrycks & Dietterich, 2019; Yin et al., 2019), Bayesian deep learning (Neal, 2012; Gal, 2016) and causality (Arjovsky et al., 2019). In this paper, we focus on the adversarial robustness where adversarial examples are carefully manipulated by human to drastically fool the machine learning models, e.g., deep neural networks, posing a serious threat especially on safety-critical applications. Currently, adversarial training (Goodfellow et al., 2014; Madry et al., 2017; Ding et al., 2018) is regarded as one promising and widely accepted strategy to address this issue.

However, similar to Out-of-Distribution (OoD) robustness, one crucial issue is that adversarial robustness also has many aspects (Hendrycks et al., 2020), mainly including *sensitivity-based robustness* (Tramèr et al., 2020), i.e. robustness against pixel-wise perturbations (normally within the constraints of an $l_p$ ball), and *spatial robustness*, i.e., robustness against multiple spatial transformations. Firstly, in the computer vision and graphics literature, there are two main factors that determine the appearance of a pictured object (Xiao et al., 2018; Szeliski, 2010): (1) the lighting and materials, and (2) geometry. Most previous adversarial robustness focus on the (1) factor (Xiao et al., 2018) based on pixel-wise perturbations, e.g., Projected Gradient Descent (PGD) attacks, assuming the underlying geometry stays the same after the adversarial perturbation. The other rising research branch tackled with the second factor, such as Flow-based (Xiao et al., 2018) and Rotation-Translation (RT)-based attacks (Engstrom et al., 2017; 2019). Secondly, by explicitly exploring the human perception, Sharif et al. (2018) pointed out that sensitivity-based robustness, i.e., $l_p$-distance

measured robustness, is not sufficient to adversarial robustness in order to maintain the perceptual similarity. This is owing to the fact that although *spatial attacks or geometric transformations* also result in small perceptual differences, they yield large $l_p$ distances.

In order to head towards the comprehensive adversarial robustness, we find that the crucial issue to investigate the aforementioned whole part of adversarial robustness is the relationships among accuracy, sensitivity-based robustness and spatial robustness. Prior to our work, a clear trade-off between sensitivity-based robustness and accuracy has been revealed by a series of works (Zhang et al., 2019; Tsipras et al., 2018; Raghunathan et al., 2020). Besides, recent work (Tramèr & Boneh, 2019; Kamath et al., 2020) exhibited that there seems to exist an obscure trade-off between Rotation-Translation and sensitivity-based robustness. However, this conclusion lacks considering Flow-based attacks (Xiao et al., 2018; Zhang & Wang, 2019), another non-negligible part in the spatial robustness evaluation, making the previous conclusion less comprehensive or reliable. As such, the comprehensive relationships among all the quantities mentioned above are still unclear and remain to be further explored. More importantly, new robust strategy that can harmonize all the considered correlations is needed, in order to achieve optimal balance within the comprehensive robustness.

In this paper, in order to design a new approach towards comprehensive robustness, we firstly explore the two main branches in the spatial robustness, i.e., Flow-based spatial attack (Xiao et al., 2018) and Rotation-Translation (RT) attack (Engstrom et al., 2019). By investigating the different impacts of these two attacks on the spatial sensitivity, we propose an integrated differentiable spatial attack framework, considering both local and global spatial vulnerability. Based on that, we present a comprehensive relationship among accuracy, sensitivity-based robustness and two branches of spatial robustness. Especially we show that the trade-off between sensitivity-based and RT robustness is fundamental trade-off as opposed to the highly interwoven correlation between sensitivity-based and Flow-based spatial robustness. We further provide strong evidence based on their different saliency maps from the perspectives of shape-bias, sparse or dense representation. Lastly, to balance these different kinds of mutual impacts within a unified adversarial training framework, we introduce the Pareto criterion (Kim & De Weck, 2005; 2006; Zeleny, 2012) in the multi-objective optimization, thus developing an optimal balance between the interplay of natural accuracy and different adversarial robustness. By additionally incorporating the two-moment term capturing the interaction between losses of accuracy and different robustness, we finally propose a bi-level optimization framework called Pareto Adversarial Training. The resulting Pareto front provides the set of optimal solutions that balance perfectly all the considered relationships, outperforming other existing strategies. Our contributions are summarized as follows:

- We propose an integrated spatial attack framework that incorporates both local and global spatial vulnerability based on Flow-based and RT attacks, paving the way towards the comprehensive spatial robustness analysis in the future.

- We present comprehensive relationships within accuracy, sensitivity-based, different spatial robustness, supported by strong evidence from the perspective of representation.

- We incorporate the Pareto criterion into adversarial robustness analysis, and are the first attempt to consider multiple adversarial robustness via the multi-objective optimization.

## 2 TOWARDS COMPREHENSIVE SPATIAL ROBUSTNESS

### 2.1 MOTIVATION

In order to better investigate the relationships between accuracy and different kinds of adversarial robustness, we need to firstly provide a fine-grained understanding of spatial robustness, which has been less studied as opposed to sensitivity-based robustness. We summarize two major branches among a flurry of related work about spatial robustness (Engstrom et al., 2017; 2019; Xiao et al., 2018; Zhang & Wang, 2019; Tramèr & Boneh, 2019; Kamath et al., 2020): (1) Flow-based Attacks, and (2) Rotation-Translation (RT) Attacks. Specifically, we find that the former mainly focuses on the local spatial vulnerability while the latter tends to capture the global spatial sensitivity. Our motivation is to firstly shed light on the fundamental difference between these two approaches, and then propose an integrated spatial robustness evaluation metric.

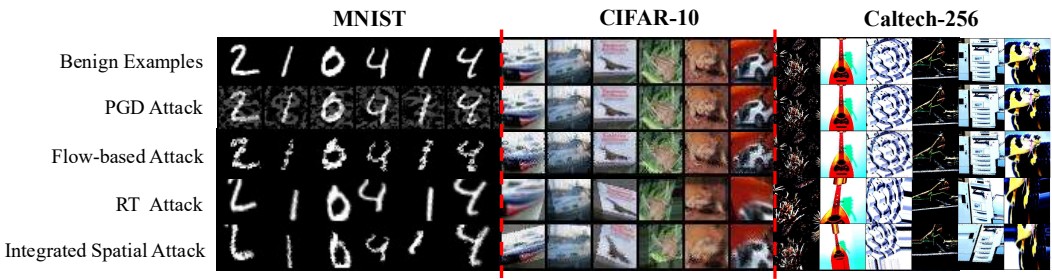

Figure 1: Visualization of Flow-based, RT and Our Integrated Spatial adversarial examples on MNIST, CIFAR-10 and Caltech-256. More discussions can refer to Appendix A.1.

## 2.2 INTEGRATED SPATIAL ATTACK: COMBING LOCAL AND GLOBAL SPATIAL SENSITIVITY

**Local Spatial Robustness: Flow-based Attacks** The most representative Flow-based Attack is Spatial Transformed Attack (Xiao et al., 2018), in which a differentiable flow vector $w_F = (\Delta\mu, \Delta v)$ is defined in the 2D coordinate $(\mu, v)$ to craft adversarial spatial transformation. The vanilla targeted Flow-based attack (Xiao et al., 2018) ($\kappa = 0$) follows the optimization manner:

$$w_F^* = \arg\min_{w_F} \max_{i \neq t} f_\theta^i(x_{w_F}) - f_\theta^t(x_{w_F}) + \tau\mathcal{L}_{flow}(w_F), \tag{1}$$

where $f_\theta(x) = \left(f_\theta^1(x), \ldots, f_\theta^K(x)\right)$ is the classifier in the $K$-classification problem. $x_{w_F}$ is Flow-based adversarial example parameterized by flow vector $w_F$. $\mathcal{L}_{flow}$ measures the local smoothness of spatial transformation further balanced by $\tau$.

Interestingly, in our empirical study shown in the left part of Figure 1, we find that Flow-based attack tends to yield local permutations among pixels in some specific regions irrespective of the option of $\tau$, rather than the global spatial transformation based on their shapes. We analyze that this phenomenon is owing to two factors: 1) Local permutations, especially in regions where colors of pixels change dramatically, are already sufficiently sensitive to manipulate, demonstrated by our empirical results shown above. 2) The optimization manner does not incorporate any sort of shape transformation information, e.g., a parameter equation of rotation, as opposed to vanilla Rotation-Translation attack, which we present in the following. Thus, Flow-based attacks tend to capture the local spatial vulnerability. Further, for the need to design the integrated spatial attack, we transform Eq 1 into its untargeted version under cross entropy loss with flow vector bounded by an $\epsilon_F$-ball:

$$w_F^* = \arg\max_{w_F} \mathcal{L}_\theta^{CE}(x_{w_F}, y) \ \ s.t. \ \|w_F\| \leq \epsilon_F \tag{2}$$

where $\mathcal{L}_\theta^{CE}(x, y) = \log \sum_j \exp\left(f_\theta^j(x)\right) - f_\theta^y(x)$. One difference compared with Eq. 1 is that we replace local smoothness term $\mathcal{L}_{flow}$ with our familiar $l_p$ constraint. Moreover, vanilla Flow-based attack (Xiao et al., 2018) follows the $max$ operation suggested in (Carlini & Wagner, 2017). However we leverage cross entropy loss instead in pursuit of a uniform optimization form in our integrated spatial attack. Proposition 1 reveals the correlation between the two loss, indicating that the smooth approximation version of $max$ operation in Eq. 1, denoted as $\mathcal{L}_\theta^S$, has a parallel updating direction with Cross Entropy loss regarding $w_F$. Proof can be found in Appendix A.2.

**Proposition 1.** *For a fixed $(x_{w_F}, y)$ and $\theta$, consider $\mathcal{L}_\theta^S(x, y) = \log \sum_{i \neq y} \exp\left(f_\theta^i(x)\right) - f_\theta^y(x)$, the smooth version loss of Eq. 1 without local smoothness term, then we have*

$$\nabla_{w_F}\mathcal{L}_\theta^{CE}(x_{w_F}, y) = r(x_{w_F}, y)\nabla_{w_F}\mathcal{L}_\theta^S(x_{w_F}, y), where \ \ r(x_{w_F}, y) = \frac{\sum_{i \neq y} \exp\left(f_\theta^i(x_{w_F})\right)}{\sum_i \exp\left(f_\theta^i(x_{w_F})\right)}. \tag{3}$$

**Global Spatial Robustness: Rotation-Translation (RT)-based Attacks** The original Rotation-Translation attack (Engstrom et al., 2017; 2019) applies parameter equations constraint on the 2D coordinate, thus capturing the global spatial information:

$$\begin{bmatrix} u' \\ v' \end{bmatrix} = \begin{bmatrix} \cos\theta & -\sin\theta \\ \sin\theta & \cos\theta \end{bmatrix} \cdot \begin{bmatrix} u \\ v \end{bmatrix} + \begin{bmatrix} \delta u \\ \delta v \end{bmatrix}. \tag{4}$$

To design a generic spatial transformation matrix that can simultaneously consider rotation, translation, cropping and scaling, we re-parameterize the transform matrix as a generic 6-dimensional affine transformation matrix, inspired by (Jaderberg et al., 2015):

$$\begin{bmatrix} u' \\ v' \end{bmatrix} = (\begin{bmatrix} 1 & 0 & 0 \\ 0 & 1 & 0 \end{bmatrix} + \begin{bmatrix} w_{RT}^{11} & w_{RT}^{12} & w_{RT}^{13} \\ w_{RT}^{21} & w_{RT}^{22} & w_{RT}^{23} \end{bmatrix}) \cdot \begin{bmatrix} u \\ v \\ 1 \end{bmatrix}, \tag{5}$$

where we denote $A_{w_{RT}}$ as the generic 6-dimensional affine transformation matrix, in which each $w_{RT}$ indicates the increment on different spatial aspects. For example, $(w_{RT}^{13}, w_{RT}^{23})$ determines the translation. Finally, the optimization form of the resulting generic and differentiable RT-based attack bounded by $\epsilon_{RT}$-ball is exhibited as:

$$w_{RT}^* = \arg\max_{w_{RT}} \mathcal{L}_\theta^{\text{CE}}(x_{w_{RT}}, y) \ \ s.t. \ \|w_{RT}\| \le \epsilon_{RT}. \tag{6}$$

**Integrated Spatial Robustness** The key to achieve this goal is to design an integrated parameterized sampling grid $\mathcal{T}_{w_{RT}, w_F}(G)$ that can warp the regular grid with both affine and flow transformation, where $G$ is the generated grid. We show our integrated approach as follows:

$$\mathcal{T}_{w_{RT}, w_F}(G) = A_{w_{RT}} \begin{bmatrix} u \\ v \\ 1 \end{bmatrix} + \begin{bmatrix} w_F \\ 1 \end{bmatrix}, \tag{7}$$

$$x^{adv} = \mathcal{T}_{w_{RT}, w_F}(G) \circ x.$$

Then we sample new $x^{adv}$ by $\mathcal{T}_{w_{RT}, w_F}(G)$ via differentiable bilinear interpolation (Jaderberg et al., 2015). Then the loss function of the differentiable integrated spatial attack can be presented as:

$$w^* = \arg\max_w \mathcal{L}_\theta^{\text{CE}}(x + \eta_w, y), \ \ s.t. \ \|w\| \le \epsilon, \tag{8}$$

where $w = [w_F, w_{RT}]^T$ and $\eta_w$ is the crafted integrated spatial perturbation. Note that $\eta_w$ itself does not necessarily satisfy the $l_p$ constraint directly. For the implementation, we follow the PGD procedure (Madry et al., 2017), a common practice in sensitivity-based attacks. We consider the infinity norm of $w$ and different learning rates for the two sorts of spatial robustness:

$$\begin{bmatrix} w_F^{t+1} \\ w_{RT}^{t+1} \end{bmatrix} = \begin{bmatrix} w_F^t \\ w_{RT}^t \end{bmatrix} + \begin{bmatrix} \alpha_F \\ \alpha_{RT} \end{bmatrix} \text{clip}_\epsilon(\text{sign}(\nabla_w \mathcal{L}_\theta^{\text{CE}}(x_{w^t}^t, y))), \tag{9}$$

$$x_{w^{t+1}}^{t+1} = \mathcal{T}_{w^{t+1}}(G) \circ x_{w^t}^t,$$

where we denote $w^{t+1} = [w_F^{t+1}, w_{RT}^{t+1}]^T$ and $\epsilon = [\epsilon_F, \epsilon_{RT}]^T$. From Figure 1, we can observe that our Integrated Spatial Attack can construct both local and global spatial transformations on images. Then, we visualize the loss surface under this Integrated Spatial Attack leveraging "filter normalization" (Li et al., 2018) as illustrated in Figure 2. It is worth noting that the highly non-concave loss landscape with respect to only rotation and translation raised by (Engstrom et al., 2019) has been largely alleviated by considering both local and global spatial vulnerability, verifying the efficiency of our Integrated Spatial Attack.

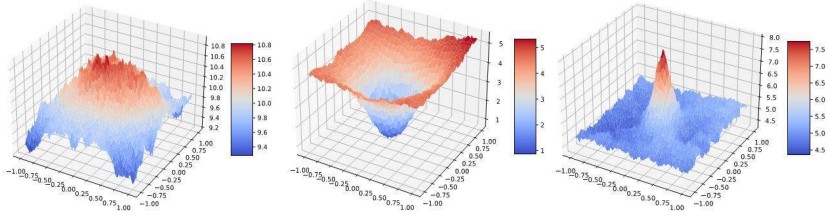

Figure 2: Loss landscape of Integrated Spatial Attack on CIFAR-10. (Left) A distant view of loss landscape w.r.t $w$ before the optimization in Eq. 8. (Middle) A close view before the optimization that shows a highly convex surface near the initialization point. (Right) The loss landscape around the maxima $w^*$ after the optimization in Eq. 8. More explanation can refer to Appendix A.3

# 3 RELATIONSHIPS BETWEEN SENSITIVITY AND SPATIAL ROBUSTNESS

## 3.1 RELATIONSHIPS

Based on the analysis above, next we focus on investigating the relationships between the strength of one specific robustness and other types of robustness. Firstly, we empirically explore these relationships through conducting thorough experiments on MNIST, CIFAR-10 and Caltech-256 datasets. By adversarially training multiple PGD (sensitivity-based) robust models with different iteration steps, we further test their Flow-based and RT-based spatial robustness via methods proposed above.

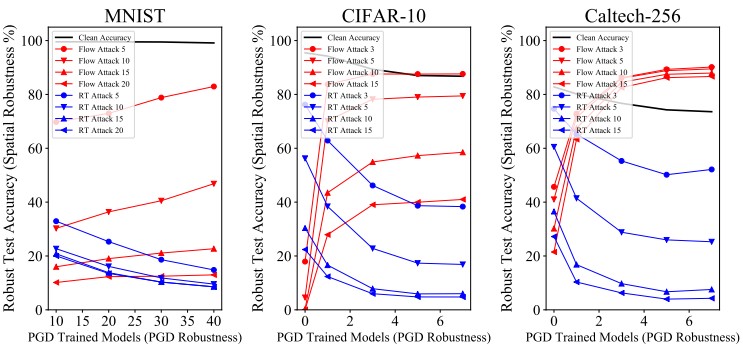

Figure 3: Relationship between sensitivity (PGD) robustness and two spatial robustness on three datasets. The X-axis represents the PGD-trained models under different PGD iterations while the Y-axis represents the robust accuracy on test data perturbed by Flow Attack (red) and RT Attack (blue).

As shown in Figure 3, it turns out that Flow-based spatial robustness (red) measured by its robust test accuracy presents a steady ascending tendency across three datasets as the PGD robustness increases, while the trend of RT-based spatial robustness fluctuates conversely. It is worth noting that we test the accuracy on correctly classified test data for the considered model for a fair comparison. The trade-off between sensitivity-based and RT-based spatial robustness is consistent with previous conclusion (Kamath et al., 2020; Tramèr & Boneh, 2019), but it does not  (even on the contrary) apply to Flow-based spatial robustness that delicately measures the local spatial sensitivity of an image. We provide the strong evidence from the perspective of representation in the next subsection.

## 3.2 EXPLANATION FROM THE VIEWPOINT OF SHAPE-BIAS REPRESENTATION

We go first with our brief conclusion: the sensitivity-based robustness corresponds to the shape-bias representation (Shi et al., 2020; Zhang & Zhu, 2019), indicating that sensitivity-based robust models rely more on global shape when making decisions rather than local texture. By contrast, the spatial robustness is associated with different representation strategies, serving as a significant supplement toward the comprehensive robust representation. To demonstrate this conclusion, we visualize the saliency maps of naturally trained, PGD, Flow-based and RT adversarially trained models on some randomly selected images on Caltech-256 exhibited in Figure 4. Specifically, visualizing the saliency maps aims at assigning a sensitivity value, sometimes also called "attribution", to show the sensitivity of the output to each pixel of an input image. Following (Shi et al., 2020;

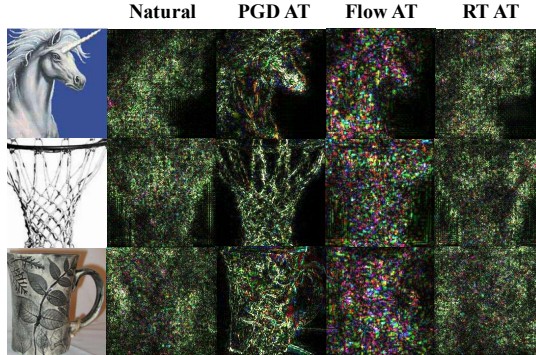

Figure 4: Saliency map of four types of training models on some randomly selected images on Caltech-256.

Zhang & Zhu, 2019), we leverage SmoothGrad (Smilkov et al., 2017) to calculate saliency map $S(x)$:

$$S(x) = \frac{1}{n} \sum_{i=1}^{n} \frac{\partial f_\theta^y(x_i)}{\partial x_i}. \tag{10}$$

Figure 4 manifests that PGD trained models tend to learn a scarce and shape-biased representation among all pixels of an image, while two types of spatially adversarially trained models suggest converse representation. In particular, the resulting representation from the Flow-based training model has the tendency towards a shape-biased one as it places extreme values on the pixels around the shape of objects, e.g., the edge between the horse and the background shown in Flow AT in Figure 4. On the contrary, RT-based models have less reliance on the shape of objects, and at the meantime, the saliency values tend to be dense, scattering around more pixels of an image. Quantitatively, we calculate the distance of saliency maps from different models across all test data on Caltech-256 dataset, and then compute their skewness shown in Figure 5.

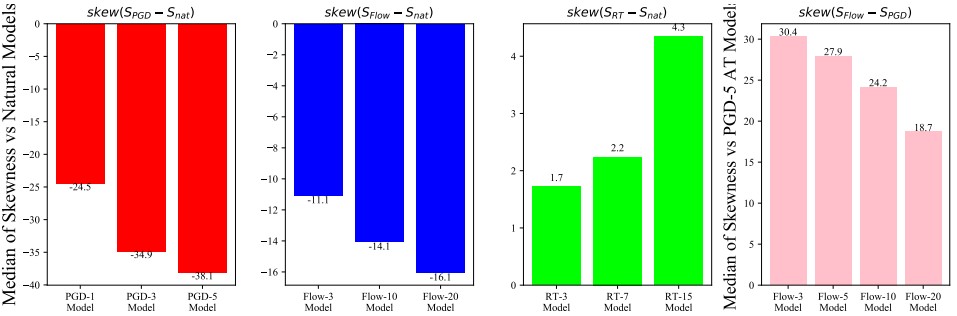

Figure 5: Median of skewness of saliency maps difference among robust models across all test data compared with other models. The first three sub-pictures are compared with the naturally trained model while the last one is compared with the PGD trained model.

Specifically, we compute the pixel-wise distance between the saliency map of a robust model and that from a considered model, and then we calculate the median of skewness of the saliency map difference among all test data. Note that if two saliency maps have no difference, then the difference values will be a normal distribution with skewness 0. A negative skewness indicates that the original saliency map (representation) tends to be sparse compared with a considered model. We plot the tendency of skewness as the strength of some specific robustness increases shown in Figure 5. Based on the observations in Figure 5, we summarize the following conclusions:

**1)** Based on the first and forth sub-pictures, both PGD and Flow-based robust models tend to learn a sparse and shape-biased representation compared with the natural model, but the Flow-based trained model is less sparse or shape-biased in comparison with the PGD trained one. **2)** On the contrary, RT-based robust models have the trend to learn a dense representation, which is also intuitive as the RT trained model is expected to *memorize* broader pixel locations to cope with potential rotation and transformation in the test data. The fundamental representation discrepancy of RT-based and sensitivity-based robustness provides deep insights to explain why the trade-off of these two robustness occurs. In the Appendix A.4, we provide a sketch map that better illustrates their relationships.

## 4 PARETO ADVERSARIAL ROBUSTNESS WITH PARETO FRONT

### 4.1 MOTIVATION

**Pareto Optimization**. Based on the aforementioned analysis on the relationships between natural accuracy and different kinds of adversarial robustness, a natural question is *how to design a training strategy that can perfectly balance their mutual impacts*, which mainly sources from their different representation manners. In particular, in most cases their relationships reveal trade-off ones, except when the sensitivity robustness increases, Flow-based spatial robustness is enhanced. To better address these competing optimization objectives, we introduce Pareto optimization (Kim & De Weck, 2005; Lin et al., 2019), and the resulting Pareto front, the set of Pareto optimal solutions, can offer valuable trade-off information between objectives. We provide more background information about Pareto optimization in Appendix A.5.

**Limitation of Existing Strategies**. Given perturbation sets $S_i, i = 1, ..., m$, and its corresponding adversarial risk $\mathcal{R}_{\text{adv}}(f; S_i) := \mathbb{E}_{(x,y)\sim\mathcal{D}}[\max_{r\in S_i} \mathcal{L}(f(x+r), y)]$, our goal is to find $f_\theta$ that can achieve the uniform risk minimization across all $S_i$ as well as the minimal risk in the natural data. **1)** Average adversarial training (Ave AT) (Tramèr & Boneh, 2019), i.e., $\mathcal{R}_{\text{ave}}(f; S) := \mathbb{E}_{(x,y)\sim\mathcal{D}}\left[\frac{1}{m}\sum_{i=1}^{m}\max_{r\in S_i}\mathcal{L}(f(x+r), y)\right]$, regards each adversarial robustness as the equal status. It may yield unsatisfactory solutions when the strength of different attacks mixed in training are not balanced, which we demonstrate in our experiments. **2)** Max adversarial training (Max AT) (Tramèr & Boneh, 2019; Maini et al., 2019), i.e., $\mathcal{R}_{\max}(f; S) := \mathbb{E}_{(x,y)\sim\mathcal{D}}[\max_i\{\max_{r\in S_i}\mathcal{L}(f(x+r), y)\}]$ may overfit to specific type of adversarial robustness if its adversarial attack used for training is too strong. Figure 6 demonstrates that as the strength of PGD attack used in Max AT increases, the comprehensive robustness of Max AT degenerates to a single PGD adversarial training, owing to the fact that the PGD loss tends to dominate among all losses. Appendix A.6 provides more details about Figure 6 and also introduces Proposition 2 to illuminate that Max AT is also closely linked with specific weights of Ave AT.

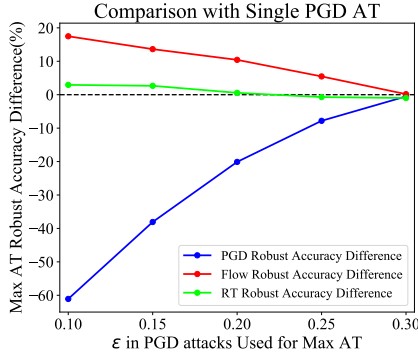

Figure 6: The difference between the model trained by the PGD method and Max AT with different parameter $\epsilon$ for the PGD attack in the adversarial training.

## 4.2 PARETO ADVERSARIAL ROBUSTNESS AND PARETO ADVERSARIAL TRAINING

The key to Pareto adversarial robustness is to find the optimal combination (trade-off in most cases) between natural accuracy, sensitivity-based and spatial robustness. Specifically, we hope to compute optimal $\alpha$ in the following formula:

$$\min_{\theta,\alpha} \mathcal{L}_\theta = \alpha_0 \mathcal{L}_{\text{nat}} + \alpha_1 \mathcal{L}_{\text{PGD}} + \alpha_2 \mathcal{L}_{\text{Flow}} + \alpha_3 \mathcal{L}_{\text{RT}}, \tag{11}$$

where $\alpha = (\alpha_0, \alpha_1, \alpha_2, \alpha_3)$ and $\mathcal{L}_\theta$ is the cross entropy loss based on the integrated framework we previously analyzed. $\mathcal{L}_{nat}, \mathcal{L}_{\text{PGD}}, \mathcal{L}_{\text{Flow}}$ and $\mathcal{L}_{\text{RT}}$ represent the natural loss, the PGD adversarial loss, the Flow-based and the RT-based adversarial loss, respectively. Note that we additionally introduce natural loss to guarantee a high-level natural accuracy (Raghunathan et al., 2020). However, direct joint minimization over Eq. 11 will degenerate to the trivial solution and the introduction of validation dataset to tune $\alpha$, e.g., DARTS (Liu et al., 2018), is also computationally expensive for the adversarial training with multiple iterations. To avoid these, our approach is to introduce the Pareto criteria to choose optimal $\alpha$ to balance the mutual impacts between different adversarial robustness. Specifically, based on Eq 11, we additionally introduce the two-moment term regarding all losses into a bi-level optimization framework, in order to compute the optimal combination $\alpha$ during the whole training process. We name this bi-level optimization approach as *Pareto Adversarial Training*, and the *lower-level* optimization regarding $\alpha$ can be formulated as follows:

$$\min_{\alpha} \sum_{i=0}^{3}\sum_{j=0}^{3}\mathbb{E}_x(\alpha_i\mathcal{L}_i - \alpha_j\mathcal{L}_j)^2 \quad \text{s.t.} \sum_{i=1}^{3}\alpha_i\mathbb{E}_x(\mathcal{L}_i) = r, \sum_{i=0}^{3}\alpha_i = 1, \alpha_i \geq 0, \forall i = 0, 1, 2, 3, \tag{12}$$

where $\mathcal{L}_0, \mathcal{L}_1, \mathcal{L}_2, \mathcal{L}_3$ represent $\mathcal{L}_{\text{nat}}, \mathcal{L}_{\text{PGD}}, \mathcal{L}_{\text{Flow}}, \mathcal{L}_{\text{RT}}$ respectively for simplicity, sharing the same model parameter $\theta$. $r$ indicates the expectation of one-moment over *all robust losses, i.e., spatial and sensitivity-based losses*, which reflects the strength of comprehensive robustness we require after solving this quadratic optimization. In particular, given the fixed $\mathbb{E}_x(\mathcal{L}_i)$ following the updating of $\theta$ based on Eq. 11, the larger $r$ we require will push the resulting $\alpha_i, i = 1, 2, 3$ larger as well, thus putting more weights on the robust losses while the whole process of Pareto Adversarial Training. For the understanding of the two-moment objective function, firstly we regard all losses as random variables with its stochasticity arising from the mini-batch sampling from data. The weighted quadratic difference is to measure the trade-off within natural accuracy and various robustness, and then the minimization is to alleviate this mutual trade-off under certain constraints. In addition, we leverage sliding windows technique to compute the expectation and *CVXOPT* tool to

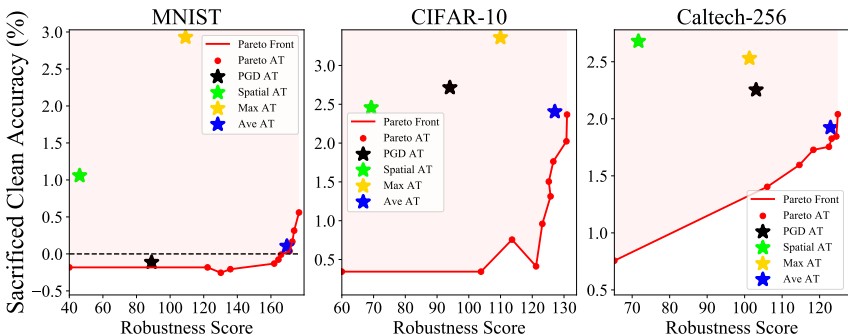

Figure 7: The Pareto front between the robustness score and sacrificed clean accuracy on MNIST, CIFAR-10 and Caltech-256. The vertical axis is the decrease of the natural accuracy compared with the naturally trained model and has been under the log transformation along two directions.

solve this quadratic optimization within each mini-batch training. Overall, for the upper level optimization in our bi-level Pareto adversarial training method, we leverage our familiar SGD method to update $\theta$ in Eq. 11 with $\alpha$ calculated from the lower level problem. In the lower level procedure, we solve the quadratic optimization regarding $\alpha$ to obtain the optimal combination among natural loss, sensitivity-based and spatial adversarial loss. We provide the proof about the quadratic formulation in Eq. 12 and our algorithm description in Appendix A.7.

### 4.3  PARETO FRONT IN EMPIRICAL STUDY

By adjusting the upper bound of expected adversarial robustness loss, i.e., $r$, we can evenly generate Pareto optimal solutions where the obtained models will have different levels of robustness under optimal combinations. The set of all Pareto optimal solutions then forms the *Pareto front*. Concretely, we train deep neural networks under different adversarial training strategies, and then evaluate their robustness by PGD, Flow-based and RT attacks, which we proposed previously, under different iteration steps. After equally averaging robust accuracy for each category of these attacks, we then compute the difference of robust accuracy between different training strategies and standard training, attaining *Robustness Score* to evaluate the comprehensive robustness of all adversarial training strategies. Finally, we plot the Robustness Score and sacrificed clean accuracy of all methods across three datasets in Figure 7. Experimental details can be found in Appendix A.8.

The Pareto criterion (Appendix A.5) exhibited in Figure 7 can be interpreted that Pareto Adversarial Training can achieve the best comprehensive robustness compared with other training strategies, given a certain level of sacrificed clean accuracy we can tolerate. By adjusting the different levels of expected comprehensive robustness $r$ in Pareto Adversarial Training, we can develop the set of Pareto optimal solutions, i.e., the Pareto front. It manifests that all other methods are above our Pareto front, thus lacking effectiveness compared with our proposal. Overall, our proposed Pareto Adversarial Training develops an optimal (Pareto) criterion, by which we can maintain the optimal balance among the mutual impacts of natural accuracy and different robustness, based on the deep understanding of their relationships.

## 5  DISCUSSION AND CONCLUSION

The essential purpose of our work is to design a novel approach towards comprehensive adversarial robustness. To achieve this goal, we firstly analyze the two main branches of spatial robustness and then integrate them into one framework. Based on that, we further investigate the thorough relationships between sensitivity-based and two distinct spatial robustness from the perspective of representation. More importantly, having understanding the mutual impacts of different kinds of adversarial robustness, we introduce Pareto criterion into adversarial training framework, yielding the Pareto Adversarial Training. The resulting Pareto front provides optimal performance under the Pareto Criterion over existing baselines. In the future, we hope to apply Pareto analysis into more general Out-of-Distribution generalization settings.

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
