# OpenReview forum: "Pareto Adversarial Robustness: Balancing Spatial Robustness and Sensitivity-based Robustness"
_ICLR.cc/2021/Conference — Reject_

### Official Review · AnonReviewer1 · 2020-10-28
**Interesting Direction, but experimental results are not clear enough**

**Rating:** 5
**Confidence:** 3

**Review:**

This paper first studies the tradeoffs between two forms of spatial robustness, including robustness against Flow-based spatial attack and Rotation-Translation (RT) attack. In particular, it proposes an approach to account for both local and global spatial transformations in an integrated framework. In addition, the paper investigates the relationship between the sensitivity-based (lp-norm based) and spatial robustness, and proposes a training method called ‘Pareto Adversarial Training’ to find optimal combination between natural accuracy, sensitivity-based and spatial robustness.


Pros:

+ Investigating the spatial robustness along with sensitivity based robustness seems to be a natural step towards comprehensive robustness, which is well motivated. Existing works on spatial robustness are well-explained.

+ Studying the saliency maps of different adversarially-trained models is interesting. Figure 4 visually looks good.

+ Introducing the Pareto optimization to account for different robustness objectives is well-motivated.



Cons:

- The experiments are the major weakness of this paper. In particular, Figure 3 is really hard to parse: (1) legends overlap with the presented curves; (2) Why is the spatial test accuracy against Flow Attack higher than clean accuracy for Caltech-256? (3) How do you set all the hyper-parameters for adversarial training and the attack strength, such as \epsilon_F in Equation 2 and \epsilon_RT in Equation 6?  (4) X-axis label says PGD Trained Models (PGD Robustness), whereas the X-axis tick represents the number of PGD attack iterations.

- Figure 7 is not explained with enough details. The X-axis label ‘Robustness Score’ is not defined throughout the paper. The meaning of each abbreviated term appeared in the legend needs clarification. Instead of a single Robustness Score, I would recommend to include the sensitivity-based robustness, Flow-based spatial robustness and RT spatial robustness individually for each models.


Other Questions and Comments:

1. From Equation 11 to Equation 12, it is unclear to me why minimizing the two-moment term regarding all losses leads to the optimal \alpha parameter. In addition, why do the constraints in Equation 12 not appear in Equation 11?

2. The presentation of the experimental sections needs to be improved in order to reach the acceptance bar of ICLR. Specifically, the experimental settings of each presented figure should be presented with enough details. Evaluation metrics of the experiments should be defined clearly in the corresponding section. Discussions on the results should be thorough enough such that each figure can be understood clearly.

---

> ### Author Response · Authors · 2020-11-16
> **Response to Reviewer 1**
>
> Here is our clarification.
>
> $\textbf{1. Experiments (1). }$
> The different tendency of the two spatial robustness is clear, even though legends may overlap with part of curves. We will improve the presentation of Figure 3 in the revised version.
>
> $\textbf{2. Experiments (2). }$
> As stated in Section 3.1 (below Figure 3), we test the robust accuracy on correctly classified test data for the considered model in order to mitigate the impact of the different generalization ability of models. Thus, a perfect robust classifier will achieve 100% robust test accuracy, which might be higher than the clean test accuracy on some datasets.
>
> $\textbf{3. Experiments (3). }$
> For the adversarial training, we strictly followed the standard settings in [1]. Please refer to Appendix A.8 for more details. As for $\epsilon_F$ and $\epsilon_{RT}$, we select a proper strength through the results of visualization in our experiments, and then adjust the attack strength mainly through changing the attack iterations.
>
> $\textbf{4. Experiments (4). }$
> X-axis label indicates the PGD adversarial trained models with corresponding number of PGD iterations.
>
> $\textbf{5. Figure 7.}$
> (1) Robustness Score: As stated on Page 8, “After equally averaging robust accuracy for each category of these attacks, we then compute the difference of robust accuracy between different training strategies and standard training, attaining Robustness Score to evaluate the comprehensive robustness of all adversarial training strategies.” Thus, Robust Score is the average robustness metric to measure all kinds of sensitivities mentioned in this paper.
> (2) We will add more clarification on the abbreviated term in the revised version.
> (3) We will provide more experimental results in the future. However, the robustness improvement in each type of corruption can be expected through our analysis in the paper. Specifically, PGD AT, Spatial AT and Max AT (as discussed on Page 7) tend to overfit to the specific type of adversarial robustness. By contrast, Ave AT and Pareto AT can achieve consistent improvement, but Pareto AT outperforms Ave AT due to the Pareto Optimality.
>
> $\textbf{6. Two-moment decision making.}$
> As discussed on Page 7, since all losses are normally competing, if one loss decreases, another loss tends to increase. Thus, we hope to minimize the mutual trade-off between each two losses, and then the two moments' decision-making framework is a natural choice. Similarly, this two-moment Pareto optimization is also applied in the optimal portfolio in finance. Also, Eq.11 is the general formulation in order to derive the more concrete lower-level constrained optimization in Eq.12.
>
> $\textbf{7. Presentation.}$
> Thank you for this suggestion. We have provided more experimental details in Appendix A.8. We promise to add more discussion about each presented figure to help readers to have a better understanding of our results.
>
> [1] Zhang, Hongyang, et al. "Theoretically principled trade-off between robustness and accuracy." ICML 2019.

---

### Official Review · AnonReviewer4 · 2020-10-28
**Official Blind Review #4**

**Rating:** 3
**Confidence:** 4

**Review:**

This work aims to study so called “comprehensive” robustness by considering natural accuracy, sensitivity-based robustness and spatial robustness simultaneously. The authors claim that they propose a “Pareto” Adversarial Training strategy to balance the mutual impact within different robustness and the resulting solutions provide the set of optimal balance among accuracy and different adversarial robustness.

Pros:
- The paper investigates comprehensive robustness. For me, the problem itself is interesting.
- This paper provides empirical evidence to show that there exists representation discrepancy for different notions of adversarial robustness. This shape-bias representation viewpoint is appealing.
- The motivation for their training strategy incorporating “Pareto criteria” is good, despite that their method has a few questionable inaccuracies summarized below.

Cons:
- The proposed “Pareto” Adversarial Training strategy is not well justified. First, the authors misuse Pareto. Pareto optimality (or front) is introduced to describe the trade-off between conflicting objectives from the multi-objective optimization. However, the proposed method is actually solving a series of bi-level optimization problems by varying the hyperparameter $r$. Since each problem is solved independently, it is not a multi-objective optimization, and the solutions obtained are also not Pareto optimal (or front). Second, in the lower-level formulation (12), the authors propose to use weighted quadratic difference to measure the trade-off. However, it is not clear why this makes sense. Is it because the two-moment objective function results in a quadratic programming problem which is easy to solve. In Section 3.2, the authors claim that there exists representation discrepancy for different adversarial robustness. Isn’t it more reasonable to consider the representation discrepancy as a measure?
- To integrate flow-based and RT-based attacks, the authors propose to use equation (2) instead of the original Flow-based attack equation (1). To justify this, the authors then provide Proposition 1 which proves that the “smooth approximation” of max operation in equation (1) has a parallel updating direction with cross entropy loss in equation (2). However, equation (1) is to minimize the max operation over $\omega_F$, whereas equation (2) maximizes the cross entropy loss over $\omega_F$. Therefore, equation (2) is different from equation (1). Moreover, the integrated spatial attack proposed in Section 2 seems not to appear anymore in their next analysis and training strategy. The readers might think that Section 2 is a little unnecessary as it is unrelated to other parts of the paper.
- In section 4.1, the authors claim that Max AT is closely linked with specific weights of Ave AT by providing Proposition 2. However, I have some concerns about the proof of Proposition 2. First, it assumes that KKT conditions hold. This assumption is strong in the sense that KKT conditions imply zero duality gap which usually does not hold for nonconvex problems. Second, even if KKT conditions hold, we can not derive that $R^*_{max}$ is a first-order stationary point of $\sum \lambda_iR^{S_i}$ because the Lagrangian function might be non-convex with resepct to $f$.

Minor comments:
- It would be better if the authors can give the definition of $x_{\omega_F}$ in (1).
- In equation (9), $clip_\epsilon$ should be used after the update of $w_F^t$.
- In equation (10), what does $n$ represent? The number of training data or the dimension of input?

---

> ### Author Response · Authors · 2020-11-16
> **Response to Reviewer 4**
>
> We respectfully disagree with the reviewer on some key points. Here is our clarification.
>
> $\textbf{1. The leverage of Pareto criteria. }$
> When we have constructed different adversarial perturbations, we are faced with the optimal coefficients to combine these adversarial losses and natural loss. These losses are normally competing with each other as we have analyzed in the former part of our paper, such as the trade-off between RT-based robustness and sensitivity-based robustness. Thus, the motivation of Pareto criteria is to find the optimal coefficients of these competing losses in order to balance all the parts of robustness. As a result, our Pareto Adversarial training is a desirable practice to incorporate Pareto criteria into the adversarial robustness. Note that since all $\alpha_i$ are computed simultaneously via a quadratic optimization, it is an obvious multi-objective optimization. Please refer to Appendix A.5 for more details.
>
> $\textbf{2. Lower-level formulation.}$
> As discussed on Page 7, since all losses are normally competing, if one loss decreases, another loss tends to increase. Thus, we hope to minimize the mutual trade-off between each two losses, and then the two moments' decision-making framework is a natural choice. Similarly, this two-moment Pareto optimization is also applied to the optimal portfolio in finance.
>
> $\textbf{3. Representation discrepancy measure.}$
> The representation of deep neural networks is still an open problem. However, recent work, such as [1], pointed out that the (PGD) adversarial training tends to form a shape-biased robust representation. This direction is very promising and can provide us with deeper insights through visualization. Inspired by this, we followed this practice and investigated the shape-biased robust representation of different adversarial training strategies.
>
> $\textbf{4. Clarification on Eq (1) and (2).}$
> We are sorry for causing your misunderstanding. Eq.(1) is to push the target logit $f_{\theta}^t$ to dominate among all classes (a larger logit margin), while Eq.(2) is to increase the loss of the current category of the data. Eq.(2) is a natural untargeted version of Eq.(1), both of which have a similar updating direction.
>
> $\textbf{5. The use of Integrated spatial attacks.}$
> As stated in the experimental details in Appendix A.8, our integrated spatial adversarial training (Spatial AT) as shown in Figure 7 is based on our proposed integrated spatial attack that unifies both Flow-based and RT-based attacks. Thus, an integrated spatial attack is necessary. Besides, when we realize the different roles of two kinds of spatial vulnerability, a natural question is whether or not we can combine the two aspects of spatial sensitivity into one spatial attack. That is the motivation of our Integrated Spatial Attack.
>
> $\textbf{6. KKT condition.}$
> Thank you for this insightful observation. Proposition 2 has revealed some correlations between Max AT and Ave AT although the conclusion can normally be made under the convex settings. We will highlight this detail in the revised version.
>
> $\textbf{7. Minor comments.}$
> (1) In the flow-based method, sample x is constructed via the flow vector $w_F$, thus we leverage $x_{w_F}$ to denote the flow-based sample. (2) Thank you for pointing out this. It is our typo. (3) Since we leverage SmoothGrad, $n$ represents the number of Gaussian noises in order to smooth the gradient.
>
> [1] Shi, Baifeng, et al. "Informative dropout for robust representation learning: A shape-bias perspective." ICML 2020.

---

### Official Review · AnonReviewer2 · 2020-10-29

**Rating:** 6
**Confidence:** 4

**Review:**

This paper first provides explanations to the inherent tradeoff between rotation adversarial attack and sensitivity attacks/spatial transform attacks, through their differences in saliency maps. Further, the authors proposed to utilize pareto training to find the best tradeoff among the four dimensions: natural accuracy, robustness against sensitivity/rotation/spatial transformation attacks. Experimental results show the proposed pareto adversarial training achieves better tradeoff between clean accuracy and adversarial robustness averaged across three types of attacks.

Pros:
1. The idea of using pareto training to tradeoff between different accuracies/robustness is interesting. This reminds me of a very recent paper [1], where the authors considered a relevant (but different) problem: How to achieve in-situ tradeoff between different attacks during inference time.
2. Using shape-bias and saliency maps to explain and demonstrate the inherent difference between rotation and sensitivity attacks is a good point.

Cons:
1. Limited novelty.
The tradeoff between rotation and sensitivity attacks has already been discussed in many previous works (e.g., [2]); the rotation and spatial translation attacks are defined in previous works [3,4] and the combination of the two is kind of trivial; the authors directly apply pareto training [5] in the adversarial tradeoff task, without any modification to fit the new problem.
2. Missing detailed experimental results.
Only the average robustness scores are provided in Figure 7. Please also consider providing robustness scores on each type of corruptions, so that we see which ones have larger improvements after pareto adversarial training and which ones do not.

[1] Once-for-All Adversarial Training: In-Situ Tradeoff between Robustness and Accuracy for Free. NeurIPS, 2020.
[2] Fundamental tradeoffs between invariance and sensitivity to adversarial perturbations. ICML, 2020.
[3] A rotation and a translation suffice: Fooling cnns with simple transformations.
[4] Spatially transformed adversarial examples. ICLR, 2018.
[5] Pareto Multi-Task Learning. NeurIPS, 2019.

---

> ### Author Response · Authors · 2020-11-16
> **Response to Reviewer 2**
>
> Thank you for your recommendation for acceptance and constructive feedback. Here is our clarification.
>
> $\textbf{1. Novelty Issue.}$
> We respectfully disagree with the reviewer.
> (a) Novelty of the trade-off relationship.
> The comprehensive relationship between spatial robustness and sensitivity-based robustness is still an open problem at least before our work. Previous work [3] only mentioned this issue in their discussion part while [6] is still unpublished, both of which only investigated the rotation-translation-based spatial perturbation (part of spatial robustness) and presented a trade-off relationship. However, [7] argued that flow-based robustness seems not to present a trade-off one, which is different from the conclusion from [3,6]. These observations motivate us to explore the comprehensive relationship further. Note that “invariance adversarial examples” defined in [2] are very different from the spatial adversarial examples we investigated here, and thus their trade-off conclusion has nothing to do with what we claimed in our paper. More importantly, through our investigation on both local and global spatial sensitivity of deep neural networks, we provide a more comprehensive relationship: the trade-off between sensitivity-based and RT robustness is fundamental, but sensitivity-based robustness is highly interwoven with flow-based spatial robustness. This conclusion provides us with deeper insights to better understand this issue.
>
> (b) Novelty of the Integrated Spatial Attack.
> The proposal of Integrated Spatial Attack is a natural outcome after analyzing the local (flow-based) and global (RT-based) spatial robustness. When we realize the different roles of two kinds of spatial vulnerability, a natural question is whether or not we can combine the two aspects of spatial sensitivity into one spatial attack. That is the motivation of our Integrated Spatial Attack. Based on our knowledge, we are the first to integrate both local and global spatial vulnerability into one attack.
>
> (c) Novelty of the Pareto Adversarial Training.
> We are not directly applying the Pareto method in [5] to the adversarial robustness setting. Specifically, [5] is a gradient-based optimization that additionally imposes restrictions on the gradients of all losses, which are very different from our bi-level two moments decision-making framework in Eq (12). Besides, Pareto optimization is a general approach in the multi-objective optimization that can obtain optimal solutions under multiple criteria. Thus, our Pareto Adversarial Training is a desirable way to incorporate Pareto criteria into solving the adversarial robustness issue, which is novel and meaningful.
>
> $\textbf{2. Experimental Details.}$
> Thank you for this suggestion and we will provide more experimental results in the appendix in the future. However, the robustness improvement in each type of corruption can be expected through our analysis in the paper. Specifically, PGD AT, Spatial AT and Max AT (as discussed on Page 7) tend to overfit to the specific type of adversarial robustness. By contrast, Ave AT and Pareto AT can achieve consistent improvement, but Pareto AT outperforms Ave AT due to the Pareto Optimality.
>
> [6] Invariance vs. Robustness Trade-Off in Neural Networks
> [7] Joint Adversarial Training: Incorporating both Spatial and Pixel Attacks

---

### Decision · Program_Chairs · 2021-01-07
**Final Decision**

**Decision:**

Reject

**Comment:**

This paper aims to address the robustness issues by considering natural accuracy, sensitivity-based robustness and spatial robustness at the same. However, the reviewers pointed out that many things, like the expriment, the presentation, the algorithm, are not clear. In addition, the technique part is weak and below the bar of ICLR.